# Development of a Real-Time 6-DOF Motion-Tracking System for Robotic Computer-Assisted Implant Surgery

**DOI:** 10.3390/s23052450

**Published:** 2023-02-22

**Authors:** Minki Sin, Jang Ho Cho, Hyukjin Lee, Kiyoung Kim, Hyun Soo Woo, Ji-Man Park

**Affiliations:** 1Department of Medical Robotics, Korea Institute of Machinery & Materials, Daegu 42994, Republic of Korea; 2Department of Prosthodontics & Dental Research Institute, Seoul National University School of Dentistry, Seoul 03080, Republic of Korea

**Keywords:** robotic computer-assisted implant surgery, dental surgery, motion-tracking system, back-drivability, Agile Eye

## Abstract

In this paper, we investigate a motion-tracking system for robotic computer-assisted implant surgery. Failure of the accurate implant positioning may result in significant problems, thus an accurate real-time motion-tracking system is crucial for avoiding these issues in computer-assisted implant surgery. Essential features of the motion-tracking system are analyzed and classified into four categories: workspace, sampling rate, accuracy, and back-drivability. Based on this analysis, requirements for each category have been derived to ensure that the motion-tracking system meets the desired performance criteria. A novel 6-DOF motion-tracking system is proposed which demonstrates high accuracy and back-drivability, making it suitable for use in computer-assisted implant surgery. The results of the experiments confirm the effectiveness of the proposed system in achieving the essential features required for a motion-tracking system in robotic computer-assisted implant surgery.

## 1. Introduction

In the field of dental surgery, high levels of accuracy are crucial for the long-term success of dental implants [1]. Failure of the accurate implant positioning can lead to problems such as damages to inferior alveolar nerve, maxillary sinus, lingual artery, etc. In order to increase implant accuracy, efforts have been made to provide detailed pretreatment planning and guidance during the surgery. Three-dimensional models of the oral anatomy generated by CT (computed tomography) scans or CBCT (cone beam computed tomography) scans have significantly contributed to the detailed planning by allowing visualization of the tissues, bones, and blood vessels as different layers [2,3]. This information can be used to guide the clinician, and is referred to as a Computer-Assisted Implant Surgery (CAIS) [4,5]. CAIS can be classified into static, dynamic, and robotic CAIS depending on how the planning information is used in the guide.

The static CAIS, which is widely accepted in the field, uses computer-guided implant surgical templates to help clinicians to locate the initial entry point of implants and decide the direction of the implants [4]. These templates are designed using software incorporating the patient’s oral and dental anatomy from CT data and have been reported to have superior accuracy compared with manual surgery in several comparative studies [6]. However, this method is difficult to cope with when the initial plan needs to be modified due to cases of poor bone quality, tissue swelling due to local anesthesia, and the presence of a bony dehiscence. The fabrication process for the surgical templates can take several hours to days, resulting in a lack of flexibility that may be considered a disadvantage. Additionally, the method may also be prone to distortion and errors due to a lack of stability in the template.

The dynamic CAIS system employs a real-time navigation system to track both the surgical tool and the target implant position [7]. This allows the clinician to perform the procedure with increased accuracy by monitoring the real-time position and orientation of the surgical tool and comparing it with the planned target position and orientation. This allows to increase overall accuracy of the procedure while maintaining simplicity in the preparation process without the need for any fabrication process. Unlike the static CAIS, the osteotomy and implant insertion can be adjusted during the surgery, but there is still a risk of human error as the method only provides visual navigation. It is also important to note that the overall accuracy highly depends on the accuracy of the real-time navigation system.


The limitations of both static and dynamic CAIS have led to the development of robotic CAIS [5,8]. The advancement in robotics technology has enabled high-precision surgery and reduced human error through real-time force guidance, thereby improving the safety of the procedure. However, to ensure high performance of the robotic CAIS, it is essential to have position registration to eliminate the difference between the position recognized by the robot and the actual position, as well as motion tracking to compensate for position errors caused by the patient’s movements. Therefore, a high-precision, real-time motion-tracking system with high bandwidth is a critical component of robotic CAIS. Currently, robotic CAIS is in its early stage of research, and not many studies have been conducted yet. Most studies use general-purpose navigation systems and have not yet focused on the potential errors that could be induced by the motion-tracking systems. Additionally, the performance specifications required of the motion-tracking system for the robotic CAIS have yet to be established.

The aim of this study is to derive the requirements for the motion-tracking system in robotic implant surgery and to propose a system to meet the requirements. The paper is organized as follows. In Section 2, the current advancements in robotic CAIS and its motion-tracking system are reviewed, including the advantages and limitations of current tracking systems and the formulation of the requirements for the motion-tracking system in implant surgery. In Section 3, a novel motion-tracking system is proposed to meet the derived requirements. Kinematics for the proposed system are analyzed in Section 4, and its resulting behaviors are presented in Section 5. Finally, the discussion and concluding remarks are available in Section 6.

## 2. Problem Formulation


In this section, the motion-tracking systems used in robotic CAIS are reviewed and their advantages and limitations are analyzed. Based on the analysis, the essential features required to improve accuracy and reliability are derived, and performance standards for the motion-tracking system are proposed.

### 2.1. State of the Art

Robotic CAIS is a newly developed technology that advances from conventional CAIS by incorporating robots to provide real-time force guidance and reduce human error during surgeries. Despite being in its early stages of development with limited studies conducted, robotic CAIS is gaining attention as a potential future technology in the field of CAIS. Robotic CAIS encompasses a range of systems, from full automation where the robot performs the surgery according to a predetermined plan to haptic guidance systems that provide the surgeon with appropriate force feedback during surgery.

The main advantage of robotic CAIS is that it can achieve higher implant placement accuracy compared with static and dynamic CAIS due to the robot’s high motion accuracy. However, in order to achieve high performance, it is important for the robot to accurately recognize the actual surgical environment. The control system of the robot must recognize the accurate distance and orientation angle between the robot base frame and the patient. Therefore, all robotic CAIS systems include a motion navigation system for measuring this relative position and motion information.

Most of the robotic CAIS systems under study primarily use optical trackers as position measurement devices. Optical trackers are vision-based measurement methods that use various 3D camera systems to determine the position and movement of reflective markers or markers with specific patterns attached to the measurement target [8,9]. This optical tracker method has the highest measurement accuracy among commercially available navigation devices, and it is highly valued for its ability to measure without physically contacting the measurement target. The spatial resolution, affected by factors such as camera resolution, marker size, and the distance between the camera and marker, is typically in the range of 20 to 200 μm [10,11,12]. However, optical trackers have several limitations. Firstly, optical methods require at least three markers for accurate measurement, making the size of the marker unit non-negligible, and occlusion issues can easily arise in cases of dental surgery with a small surgical area. Secondly, this method is highly sensitive to lightning conditions in the measurement environment, and it has a limitation in fast motion measurement due to its low sampling rate. Finally, measuring a new target requires calibration, making it inconvenient to use.

Another motion navigation method is a physical measuring device in the form of a robotic arm, known as a Coordinate Measuring Machine (CMM) [13]. This method measures the position and movement of the target through the posture of the robotic arm by physically contacting the object. It has high accuracy, reliability, and repeatability, as well as a high sampling rate. Furthermore, after the position calibration, it can be repeatedly used for multiple targets, leading to high usability. The Yomi Robotic System uses the CMM-type motion navigation system, known as the Patient Tracking Arm [14]. At present, the Yomi Robotic System is the only commercially available robotic CAIS system, and its use of this method is expected due to its reliability and usability. However, CMM-type tracking system has a fundamental limitation that the weight and inertia of the system can affect the movement of the measurement target. Therefore, it is important for CMM-based measuring devices to have lightweight and high dynamic back-drivability characteristics to be used as CAIS. To the best of the authors’ knowledge, there is no detailed explanation of the measurement system used in the Yomi, but we can anticipate significant improvements in dynamic back-drivability.

### 2.2. Requirements

Based on the analysis from the previous subsection, commercially available navigation systems have limitations to be employed in CAIS. In this subsection, we discuss the essential features that must be implemented in a navigation system for CAIS. Four factors have been identified as crucial, and a precise objective has been set for each issue.

#### 2.2.1. Workspace

In order to track the patient’s movement during implant surgery, it is important to define the patient’s head’s range of motion (ROM). A human head without any restraint has a ROM of −70 to 70${}^{\circ }$ for left/right rotation, −55 to 50${}^{\circ }$ for cervical flexion/extension, and −40 to 40${}^{\circ }$ for left/right lateralization [15]. However, in the implant surgical condition, it may not be necessary to implement this full ROM, because the patient is lying on the dental chair, and the contact between the patient’s back and head with the chair significantly restricts movement. In addition, head movements are limited to cases where a clinician adjusts the angle of the head to improve the field of view and unintentional reflex movements of the patient.

To consider surgical environment, we make some hypotheses to set the target ROM. First, we attempt to identify the rotation center for the head movements. As shown in Figure 1d, we assume that left/right head rotation occurs around the center of the neck/ Since the patient’s back and head are in contact with the dental chair, we assume the motions are severely restricted to −30 to 30${}^{\circ }$ of left/right rotation, −20 to 20${}^{\circ }$ of cervical flexion/extension, and −15 to 15${}^{\circ }$ of left/right lateralization. Figure 1 shows the target ROM determined under these hypotheses and the nominal full ROM of the head.

#### 2.2.2. Sampling Rate

Dynamic motions are more difficult to capture than static or slow movements. Nyquist–Shannon suggests at least double the sampling rate of maximum motion frequency to capture its motions. However, even a hundred Hz may not be sufficient for some specific cases [16]. Motion capturing devices used in sports applications commonly use a sampling rate between 50 and 250 Hz [17]. Sampling rates over 1 kHz are only necessary for some specific cases, including impact or very high-velocity movements.

In robotic CAIS, however, captured motions are not only to be stored but to be used as a target position to compensate robot posture. A higher sampling rate reduces delays in transmitting patient posture information to the robot. Therefore, the goal is to achieve the highest sampling rate, which is often over 1 kHz.

#### 2.2.3. Accuracy

To evaluate the accuracy of dental implant placement, the common variables to be measured include angular deviation, global coronal deviation (error at the implant top point), and global apical deviation (error at the implant apex point) [2]. Since the apical deviation is affected by the direction of the angular deviation, the absolute value may not provide a suitable comparison between studies. Therefore, only the coronal and angular deviations are used for comparison.

As summarized in Table 1, the conventional free-hand method results in average errors of 1.62 to 1.93 mm and 5.85 to 8.7${}^{\circ }$ for coronal deviation and angular deviation, respectively. Recent methods with CAIS greatly improve the outcome with the use of surgical template, optical tracker, and YOMI dental robotic systems, as shown in Table 1.

There are various factors that can cause these errors, including human error by the clinician. However, it is clear that CAIS with the navigation systems greatly facilitates to increase the overall accuracy of dental implant placement. The higher the surgical navigation system’s resolution, the more it improves the accuracy of the robot-assisted surgical navigation system. The highest resolution commercially available for the navigation system may be around 100 μm, using an optical tracker [10]. Therefore, the goal is to achieve the same or better resolution, meaning less than 100 μm.

#### 2.2.4. Back-Drivability

Back-drivability refers to the amount of resistive force or torque required to be moved. Ideally, the device should not exert additional force on the operator during movements, which requires low friction and low inertia. If the motion tracker lacks back-drivability, resistive forces may cause problems, such as alveolar bone damage or splint deformation. As discussed in the previous subsection, conventional CMM is not suitable due to its low back-drivability performance, mainly due to the use of reducer that magnifies frictional and inertial forces.

There are high back-drivable mechanical devices, such as Phantom (3D systems, Rock Hill, SC, USA) [22], Delta (Force dimension, Nyon, Switzerland) [23], etc. The back-drive friction is reported as 0.1 to 0.7 N for these devices [22,24]. Designing for high back-drivability involves limiting the use of reducers (such as gear mechanisms), using a light-weight link structure, and incorporating gravity compensation. The goal of this study is to implement a back-drivable force at an end-effector of 0.1 N or less, which is comparable to the highest level of available mechanical devices.

## 3. Methods

There have been many studies using optical systems as a motion-tracking system in robotic surgery. As discussed in the previous section, an optical system has the advantage of high accuracy and noncontact measurement, but the sampling rate may not be enough to respond to fast motions and difficult-to-handle issues such as marker fixation, light source, and obstruction of vision. To overcome these problems, we propose a motion-tracking system based on the CMM method. Despite the inherent limitations of the CMM method, we designed a novel 6-DOF motion-tracking system that meets our requirements for real-time measurements and back-drivability performance. The total system consists of an end-effector splint, a 3-DOF translational motion-tracking structure, and a 3-DOF rotational motion-tracking structure. The structures for tracking translational and orientation motions are decomposed to minimize the coupling effect between each motion.

### 3.1. Splint

The dental splint is a component that connects the motion-tracking device to the patient. It is designed based on the splint used in the Yomi Robotic System [21]. The Edentulous Patient Splint (EPS) is a method that fixes the splint securely to the patient’s alveolar bone using monocortical bone screws. This method is invasive, but since the design of the splint is not the main focus of this study, the most rigid connection method was selected to minimize errors caused by the splint.

The surgical splint is fabricated with a Stereo Lithography Apparatus (SLA) 3D printer (Objet30, Stratasys, Eden Prairie, MN, USA), as shown in Figure 2. It consists of a fixation part that connects to the patient’s alveolar bone and a connector part for combining with the motion tracker arm. The upper surface of the connector part has three CT markers for position registering the target implant position and the center of the CT markers. There is a tap hole in the center of the CT marker for assembling the splint and the end-effector of the motion tracker arm.

Figure 3 shows the simulated results of the end-effector trajectories for the given target ROM of the patient’s movement. As illustrated in Figure 3a, the red rod stands for the end-effector, which may include the dental splint and final link of the tracking system, and it is attached to the patient. The yellow dots represent the range of the target implant insertion points, which are assumed to be centered at the front teeth. The blue dot and ellipsoid are numerically computed to indicate the target workspace the motion-tracking system should implement. The reachable workspace for the end-effector must have dimensions larger than 332×174×82 mm${}^{3}$ (width × height × length).

### 3.2. Translational Motion-Tracking Structure

The translational motion-tracking structure is designed as a 3-DOF serial linkage mechanism that implements the XYZ translational motion of the end-effector. The structure is designed to be light and compact to reduce the inertial effect while satisfying the required workspace.As shown in Figure 3, the target workspace has a wide range in the XY plane and a relatively small range in the Z direction. To achieve this workspace, the 3-DOF linkage structure is designed to have two yaw joints (Joint 1 and 3 in Figure 4) and one pitch joint (Joint 2 in Figure 4). The parallelogram structure between joints 2 and 3 ensures that the axes of rotation of joints 1 and 3 are always kept parallel, allowing joint 3 to always move on a horizontal plane. Moreover, this configuration makes it easy to implement weight compensation, because the potential energy change occurs only with a single joint motion (Joint 2).

The lengths of the components in the 3-DOF translational motion-tracking structure are determined based on the target workspace. The length of the parallelogram $l_{1}$, which provides independent vertical displacement, is set to be 120 mm. The lengths of $d_{1}$ and $d_{2}$ are designed to be 30 mm and 34 mm, respectively, to avoid interference with the encoders at joints 1 and 3. The lengths of $l_{2}$ and $l_{3}$ are designed to be 120 mm and 40 mm, respectively, to meet the target horizontal workspace. There is an adjustable locking joint that can be manually adjusted, and in this study, it is fixed at an angle of $60^{\circ }$.

### 3.3. Rotational Motion-Tracking Structure

The rotational motion-tracking structure should be able to implement three angular motions of roll, pitch, and yaw with respect to one rotational center. We considered two mechanical structures to achieve this capability.

The first structure is an articulated link mechanism, as depicted in Figure 5. It comprises three revolute joints and two $90^{\circ }$ bent linkages with the three rotation axes orthogonally to meet at one point in the initial configuration. The simple design and ease of manufacturing with high mechanical tolerances make this structure advantageous. However, this structure has unequal moment arm lengths for each rotation motion, causing anisotropic interaction forces due to the device’s inertia. Moreover, the wires connected to encoders on each rotating axis can cause unwanted torques and joint stiffness during operation. These two issues result in discomfort during rotational motions. Furthermore, a singularity issue arises when the second joint rotates $90^{\circ }$, making the first and third joints become collinear, which limits the ROM of the second joint to prevent this issue.

The second mechanism is a spherical parallel structure called the Agile Eye mechanism [25]. The Agile Eye mechanism is a special configuration of a 3-RRR spherical parallel mechanism. As shown in Figure 6, it has three legs connecting a moving platform to a fixed base and can calculate three rotation angles of the end-effector through the base’s three rotation axes. The Agile Eye structure has the advantage of having an encoder in a fixed position, which prevents unnecessary joint stiffness from encoder wires. Additionally, It has an isotropic structure, meaning that it has the same interaction force for movement in each direction. Depending on design conditions, the Agile Eye structure can prevent singularities from occurring. However, its complex structure makes it difficult to maintain high mechanical tolerance and joint stiffness.

In conclusion, after evaluating the characteristics of each structure, it is determined that the Agile Eye structure is the more suitable choice for use as a motion-tracking device. The designed Agile Eye structure, shown on the right side of Figure 6, provides rotational motion tracking with a range of $\pm 60^{\circ }$ pitch and yaw and $\pm 30^{\circ }$ roll without singularity.

### 3.4. Sensor

The proposed mechanism has a maximum distance between the origin and the rotational center of the Agile Eye, which is approximately 370 mm. To achieve a target resolution of 100 μm, the encoder resolution must be better than 0.00027${}^{\circ }$/tick, which requires 17-bit resolution. In order to maximize the back-drivability performance, the use of a reduction drive mechanism should be avoided. As a result, the proposed motion-tracking system uses a high-resolution 18-bit encoder (EBI1135, Heidenhain GmbH, Traunreut, Germany) for all active joints. The size of the encoder is ϕ 37.0 mm with a height of 13.0 mm.

### 3.5. Gravity Compensation

To enhance the back-drivability, a passive gravity compensation mechanism is employed in the motion tracker. The passive gravity compensation mechanism that does not require any actuator is employed to maximize the back-drivability. The gravity compensation is implemented separately for each translational and rotational motion-tracking structure to enable complete gravity compensation of the proposed tracker.

For the translational motion-tracking structure, a spring-based weight compensation mechanism is used, as shown in Figure 7a. This mechanism creates a weightless condition by balancing the sum of the potential energy of the linkage structure and the potential energy stored in the spring. A structure using 1-DOF linkage and an ideal spring with zero-initial-length characteristic is widely used, however, actual springs do not have this characteristic, so a pulley–wire mechanism is employed to simulate the ideal spring [26,27]. This method has the advantage of not increasing the overall weight of the system, however, the stiffness of the spring may affect the natural frequency of the system and hinder its dynamic motion. The spring coefficient for the weight compensation, *k*, can be computed as $k=\frac{Mgl_{1}}{d_{3}d_{4}}$, where $d_{3}$ is the offset length between the encoder and the pulley, $d_{4}$ is the length from the pulley to the routing point, *M* is the combined mass beyond the parallelogram structure, and *g* is the gravitational acceleration.

For the rotational motion-tracking structure, a counterweight-based method is used for weight compensation. Gravity compensation using a counterweight appears to be the simplest method to position the counterweight on the opposite side of the system’s center of mass. Although this approach increases the system’s total mass and inertial effects, it does not alter the mechanical stiffness and bandwidth of the system. However, in our case, the impact of these disadvantages is small because the end-effector only has light parts, such as a splint and a rod, making the required counterweight light. The implementation of the counterweight in the rotational motion-tracking structure is depicted in Figure 7b.

## 4. Kinematics

In this section, we formulate kinematic equations to obtain end-effector position and orientation. For the proposed mechanism, some joint angles may not be directly accessible. Hence, the geometric relationship between each joints is essential to explore kinematics of the robots. This also requires discussing singularity conditions of the link structure.

Rigid-body frames are attached to the proposed device, as depicted in Figure 8. $F_{0}$ represents the world coordinate as the base frame. The frames $F_{1}$ to $F_{10}$ are assigned from the base of the device to the final tip of the end-effector. The unit vectors for $F_{i}$ are denoted by $x_{i}$, $y_{i}$, and $z_{i}$. The relative configuration of a moving frame *A* relative to a fixed frame *B* is represented by the homogeneous transformation matrix, denoted as
(1)$${}^{B}T_{A}=\left[\begin{array}{cc} {}^{B}R_{A} & {}^{B}d_{A}\\ 0_{1\times 3} & 1, \end{array}\right]$$
where a rotation matrix ${}^{B}R_{A}\in SO\left(3\right)$ and a displacement vector ${}^{B}d_{A}\in R^{3}$.

As shown in Figure 8, the proposed mechanism consists of two main parts: a 3-DOF articulated link structure and a 3-DOF spherical parallel link structure. Based on the assigned frame for the articulated link structure, $F_{1}$ to $F_{5}$, Denavit–Hartenberg parameters can be found in Table 2. This facilitates homogeneous transformations from $F_{i}$ to $F_{i-1}$ as follows:(2)$${}^{i-1}T_{i}=\left[\begin{array}{cccc} \cos q_{i} & -\sin q_{i}\cos\alpha_{i} & \sin q_{i}\sin\alpha_{i} & a_{i}\cos q_{i}\\ \sin q_{i} & \cos q_{i}\cos\alpha_{i} & -\cos q_{i}\sin\alpha_{i} & a_{i}\sin q_{i}\\ 0 & \sin\alpha_{i} & \cos\alpha_{i} & d_{i}\\ 0 & 0 & 0 & 1 \end{array}\right].$$

The given variables in Table 2 can be obtained from link lengths which are $d_{1}=30$ mm, $l_{1}=120$ mm, $d_{2}=34$ mm, $l_{2}=120$ mm, and $l_{3}=104.71$ mm (Here, $l_{3}$ includes the distance to the rotational center of the Agile Eye, so it differs from the number suggested in Section 3), and $h_{1}=48.25$ mm.

The Agile Eye mechanism, however, does not allow direct access for some joint angles, unlike the articulated link structure. Hence, Denavit–Hartenberg parameters are no longer to be used for homogeneous transformations. Since the kinematic solution of the Agile Eye is not trivial, numerous studies have been conducted, such as [28] and references therein. Among the various solutions and methods, we may follow the concept and notations used in [29]. Here, we assign frames $F_{6}$ to $F_{8}$ to facilitate computations of its homogeneous transformation matrix. The origin of $F_{5}$ is the rotation center of the Agile Eye and is shared as the origins of $F_{6}$ to $F_{8}$.

The unit vectors of the base frame for the Agile Eye in [29] are defined along the axis of each active joints. Therefore, it is required to define $F_{6}$ to match the definition of the base frame of the Agile Eye. As illustrated in Figure 9, the homogeneous transformation matrix from $F_{6}$ to $F_{5}$ can be written as
(3)$${}^{5}T_{6}=\left[\begin{array}{cccc} -\sqrt{\frac{1}{3}} & -\sqrt{\frac{1}{3}} & -\sqrt{\frac{1}{3}} & 0\\ 0 & \sqrt{\frac{1}{2}} & -\sqrt{\frac{1}{2}} & 0\\ \sqrt{\frac{2}{3}} & -\sqrt{\frac{1}{6}} & -\sqrt{\frac{1}{6}} & 0\\ 0 & 0 & 0 & 1 \end{array}\right].$$

$q_{4}$, $q_{5}$, and $q_{6}$, which are the active joint angles, correspond to the unit vectors $x_{6}$, $y_{6}$, and $z_{6}$, respectively. Then, the homogeneous transformation matrix describing the moving platform of the Agile Eye can be formulated as ${}^{6}T_{7}=\left[\begin{array}{cc} {}^{6}R_{7} & 0_{3\times 1}\\ 0_{1\times 3} & 1 \end{array}\right]$, where
(4)$${}^{6}R_{7}=\left[\begin{array}{ccc} 1 & 0 & 0\\ 0 & \cos q_{4} & -\sin q_{4}\\ 0 & \sin q_{4} & \cos q_{4} \end{array}\right]\left[\begin{array}{ccc} \cos\phi_{3} & -\sin\phi_{3} & 0\\ \sin\phi_{3} & \cos\phi_{3} & 0\\ 0 & 0 & 1 \end{array}\right]\left[\begin{array}{ccc} \cos\phi_{2} & 0 & \sin\phi_{2}\\ 0 & 1 & 0\\ -\sin\phi_{2} & 0 & \cos\phi_{2} \end{array}\right]$$
and $\phi_{3}$ and $\phi_{2}$ denote passive joint angles of the leg 1, as defined in [29]. With the inherent geometric relationship from the kinematic chain of the mechanism, one can derive the constraint equations of the Agile Eye such that
(5)$$\left\{\begin{array}{c} a_{1}\cos\phi_{2}+b_{1}\sin\phi_{2}=0\\ a_{2}\cos\phi_{2}+b_{2}\sin\phi_{2}=0 \end{array}\right.$$
where, $a_{1}=\sin q_{5}\cos q_{4}$, $b_{1}=-\cos\phi_{3}\cos q_{5}+\sin q_{5}\sin q_{4}\sin\phi_{3}$, $a_{2}=\cos\phi_{3}\sin q_{6}-\cos q_{6}\cos q_{4}\sin\phi_{3}$, and $b_{2}=-\cos q_{6}\sin q_{4}$. At this point, it is important to note that we impose several assumptions on the Agile Eye to avoid singularity of the mechanism. Each leg does not reach its boundaries to avoid Type 1 singularity condition. With this assumption, the working mode of the Agile Eye remains the same as the reference configuration. The Type 2 singularity (only happens in parallel mechanism) condition can be obtained by differentiating the constraint equations, which are followed by
(6)$$\sin q_{4}\sin q_{5}\sin q_{6}+\cos q_{4}\cos q_{5}\cos q_{6}=0.$$

Therefore, we assume that $\sin q_{4}\sin q_{5}\sin q_{6}+\cos q_{4}\cos q_{5}\cos q_{6}\neq 0$. These two assumptions may greatly simplify the original solutions to obtain $\phi_{3}$ and $\phi_{2}$ from $q_{5}$ and $q_{6}$ such that
(7)$$\left\{\begin{array}{cc} \cos\phi_{3}= & \frac{abs(c_{2})}{{\left(c_{1}^{2}+c_{2}^{2}\right)}^{1/2}}\\ \sin\phi_{3}= & -\frac{c_{1}}{c_{2}}\cos\phi_{3}\\ \cos\phi_{2}= & sign\left(\cos q_{6}\right)\frac{abs(b_{1})}{{\left(a_{1}^{2}+b_{1}^{2}\right)}^{1/2}}\\ \sin\phi_{2}= & -\frac{a_{1}}{b_{1}}\cos\phi_{2} \end{array}\right.$$
where
$$\left\{\begin{array}{c} c_{1}=\sin q_{5}\cos q_{4}\cos q_{6}\sin q_{4}-\sin q_{6}\cos q_{5}\\ c_{2}=\sin q_{4}\sin q_{5}\sin q_{6}+\cos q_{4}\cos q_{5}\cos q_{6} \end{array}\right.$$
and more details about the formulation can also be found in [29] and references therein.

For convenience, we introduce another frame $F_{8}$, which is identical with $F_{5}$ in reference configuration. Then, the homogeneous transformation matrix ${}^{7}T_{8}$ is equal to ${}^{5}T_{6}^{⊤}$. Finally, if we denote the tip position as ${}^{8}d_{9}$, the homogeneous transformation matrix for the tip is represented as ${}^{8}T_{9}=\left[\begin{array}{cc} I_{3\times 3} & {}^{8}d_{9}\\ 0_{1\times 3} & 1 \end{array}\right]$. Thus, the forward kinematics for the end-effector in world coordinate can be formulated in terms of homogeneous transforms such as ${}^{0}T_{9}$.

## 5. Results

Figure 10 depicts the 6-DOF motion-tracking system that was developed and attached to a dental model (Dentium, Republic of Korea). In this section, we conduct analyses and experiments to assess the proposed requirements of the developed system.

### 5.1. Workspace

The workspace achievable by the developed motion-tracking system was numerically computed to determine if it meets the target workspace. Figure 11a shows the Cartesian workspace with fixed orientation at reference configuration. The blue and red dot lines represent the link structure for translational motion and the auxiliary line to the rotation center of the Agile Eye, respectively. The transparent blue spheres represent the reachable points by the rotation center point of the Agile Eye, while the gray area is the required workspace. The yellow area in Figure 11b represents the workspace that can be reached by the developed tracking systems, which is 1.8 times larger than the objective.

Figure 11c,d shows the entire motion-tracking system including the patient. The newly added red solid lines represent the end-effector of the tracker from the rotation center of the Agile Eye. Figure 11c provides an insightful illustration of the tracker posture for a certain point within the yellow ellipsoid. The rotation angle changes are limited within $\pm 60^{\circ }$ for pitch and yaw motions and $\pm 30^{\circ }$ for roll motion, as designed.

### 5.2. Sampling Rate

An associated sensor interface hardware was specifically developed. The main control unit with a microcontroller (STM32F767, STMicroelectronics, Geneva, Switzerland) directly receives the EnDat22 signal from six encoders and computes forward kinematics to obtain several marker positions. These computed marker position values can be transmitted to the external computer at 1 kHz via real-time Ethernet communications (UDP). Additionally, a real-time control platform (Performance real-time target machine, Speedgoat GmbH, Bern, Switzerland) with MATLAB Simulink (R2020b, Mathworks Inc., Natick, MA, USA) was used to receive real-time information and record experimental data.

### 5.3. Accuracy

An experiment was designed to evaluate the positional accuracy of the developed motion-tracking system. The performance of the motion tracker was evaluated by determining the RMS value and standard deviation (STD) of the positions measured during 30 repeated movements between the origin and the target position. Since the developed system does not include any actuator to create motions by itself, a robot system was used to generate repetitive motion. A 3-DOF robot system capable of realizing translational movement in three directions of the *XYZ* axis was used, and the end-effector of the robot and the end-effector of the motion tracker were rigidly attached through a connector block, as shown in Figure 12. Additionally, a laser tracker (Leica Absolute Tracker AT960, Leica Geosystems, Heerbrugg, Switzerland) with a measurement accuracy of 20 μm was used to measure reference data to verify whether the experiment process was properly performed. The laser tracker was installed at a distance of about 2 m from the connector block, and the Spherical Mounted Retroreflector (SMR) was attached at the connector block as a marker for the laser tracker. The entire experimental setup is shown in Figure 12.

The position data using the motion-tracking system were measured at a sampling rate of 1 kHz, and the results of repeatedly moving between two points are shown in Figure 13. Figure 13a illustrates the Cartesian position data obtained from the developed motion tracker, while Figure 13b depicts the overlaid spatial position at the sampled times. The RMS and STD of the position accuracy calculated based on these results are shown in Figure 14. The RMS value of the position measured using the developed motion-tracking system was 68.2 μm, and the STD value was 35.9 μm. The laser tracker gave 34.8 μm for the RMS value and 17.8 μm for the STD value.

### 5.4. Back-Drivability

An experiment was designed to evaluate the back-drivability of the developed motion-tracking device. Frictional and inertial forces generated during arbitrary human motions are measured at the tip of the motion-tracking system using a 6-axis F/T sensor (Mini45, ATI Industrial Automation, Apex, NC, USA) attached at the tip of the motion tracker arm, as shown in Figure 15. The range of the F/T sensor are ±290 N for *x* and *y* axis, ±580 N for *z* axis. The resolution is reported as $\frac{1}{8}$ N which is typical for most applications.

The F/T data can be measured at the rate of 1 kHz using a DAQ board (IO102, Speedgoat GmbH, Switzerland) and a real-time control platform (Performance real-time target machine, Speedgoat GmbH, Switzerland) with MATLAB Simulink (R2020b, Mathworks Inc., Natick, MA, USA). The arbitrary motions captured by the developed motion tracker are shown in Figure 16a,b, where the red line and black circles represent the trajectory and sampled points taken every 0.5 s, respectively. Figure 16c shows the measured force data according to arbitrary human motions. The RMS values of the interaction forces calculated for the axes *X*, *Y*, and *Z* are 0.207 N, 0.465 N, and 0.274 N, respectively.

### 5.5. Discussion

The proposed motion-tracking system meets most of the requirements for being used as a tracking system in robotic CAIS. However, we would like to highlight current limitations and discuss how to improve the current device.

The derived workspace, as shown in Figure 11, has the shape of an eggshell, with a wide range of motions in the lateral direction but a narrow range in the longitudinal direction.
This means that while the total workspace of the motion tracker is larger than the target workspace, its limited longitudinal displacement requires that the base position be carefully chosen to cover the entire target workspace. Improving factors, such as link length or the ratio of linkages, could increase the tracker’s workspace in the vicinity of the target workspace, but it is important to balance the size of the system with the feasible workspace.

The results of the accuracy experiment indicate that the developed motion-tracking system met the required accuracy performance. In the experiment to measure accuracy, a robot system was utilized for repetitive movements. The reference data of 34.8 μm measured by the laser tracker reflect the positioning accuracy of the robot system. However, this positioning accuracy of the robot was deemed insufficient for evaluating the performance of the motion-tracking device, as it may result in the perceived accuracy of the motion tracker being lower than its actual performance. Future verification with a more precise position input device is necessary to establish the true accuracy of the motion tracker. Although the developed device satisfied the required performance, it showed insufficient performance compared with the laser tracker. These accuracy errors can also be caused by structural deformations of joints and linkages, so it is necessary to improve the system rigidity through structural improvements and material upgrades. Additionally, since there is currently no calibration method for evaluating and compensating the measurement accuracy of each sensor, the measurement error of the sensor may affect the accuracy, so in the future, an appropriate calibration method for the device should also be established.

The back-drivability performance of the motion tracker was evaluated based on the RMS value of the interaction forces, which was found to be between 0.21 to 0.47 N, which is insufficient compared with the target value of 0.1 N. The nonredundant 6-DOF configuration of the proposed motion tracker ensures repeatability during movement, but it resulted in significant inertial forces due to the large distance the device must move for even slight head movements. To improve the back-drivability and reduce interaction force, incorporating a redundant mechanism to minimize joint movements and reduce joint friction within the system may be necessary. Furthermore, measurement noise in the F/T sensor data may contribute to the results, and adequate signal filtering should be used to solve this issue.

## 6. Concluding Remarks

In this work, we analyzed the workspace, sampling rate, accuracy, and back-drivability of the proposed motion-tracking system for use in robotic CAIS. We proposed the motion-tracking system based on the CMM method, which is expected to provide real-time measurement and high back-drivability performance, with higher reliability and usability compared with optical tracking methods commonly used in robotic CAIS studies.


The proposed motion-tracking system was found to have satisfactory results in most aspects, but with insufficient back-drivability performance, as determined through experiments. The back-drivability issue is a fundamental problem in the CMM method, as the equipment’s inertia cannot be zero, resulting in resistance during operation. This problem becomes more severe if the required workspace increases and the device become larger. Although the current design has been proposed as an initial suggestion, it has not been optimized for dental applications yet. Additional research is required to develop a mechanism that can efficiently cover the required workspace with minimal movement.

We believe that the proposed requirements and the device have the potential to contribute to the field of robotic CAIS. In the near future, we are going to improve the current system and integrate it with the robotic CAIS systems for clinical studies.

## Figures and Tables

**Figure 1 sensors-23-02450-f001:**
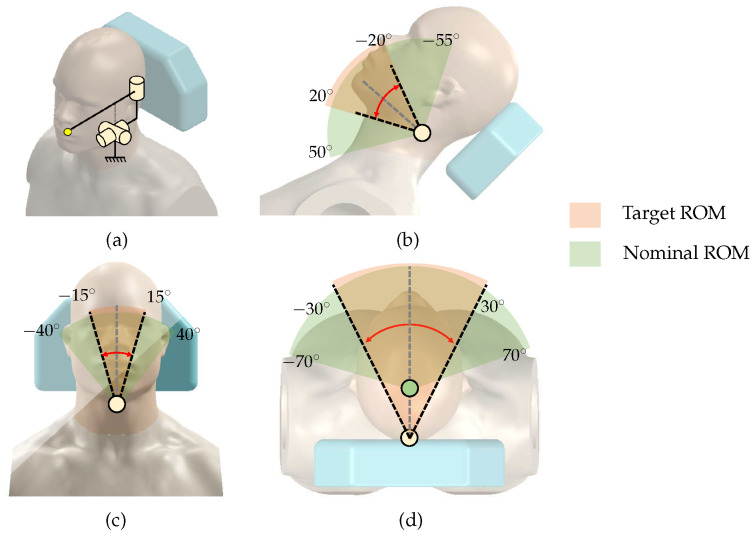
Head range of motion: (**a**) head–neck movement equivalent model; (**b**) cervical flexion/extension; (**c**) left/right lateralization; (**d**) left/right rotation.

**Figure 2 sensors-23-02450-f002:**
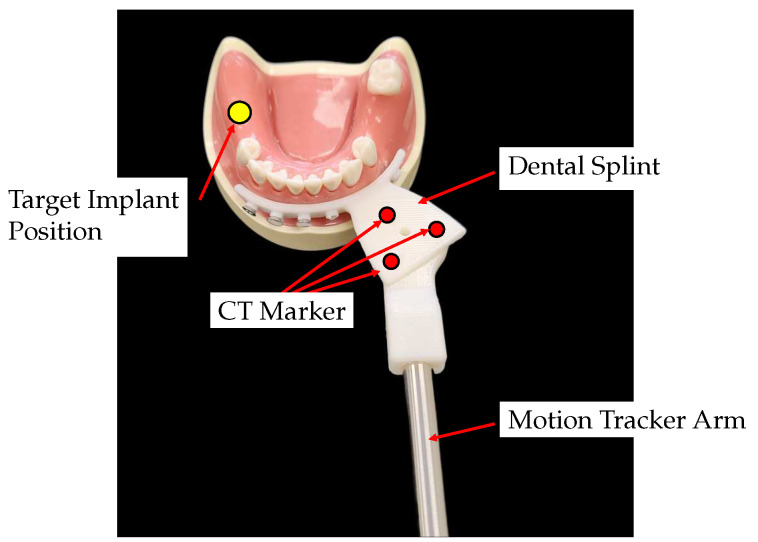
The dental splint to connect the motion tracker and the patient.

**Figure 3 sensors-23-02450-f003:**
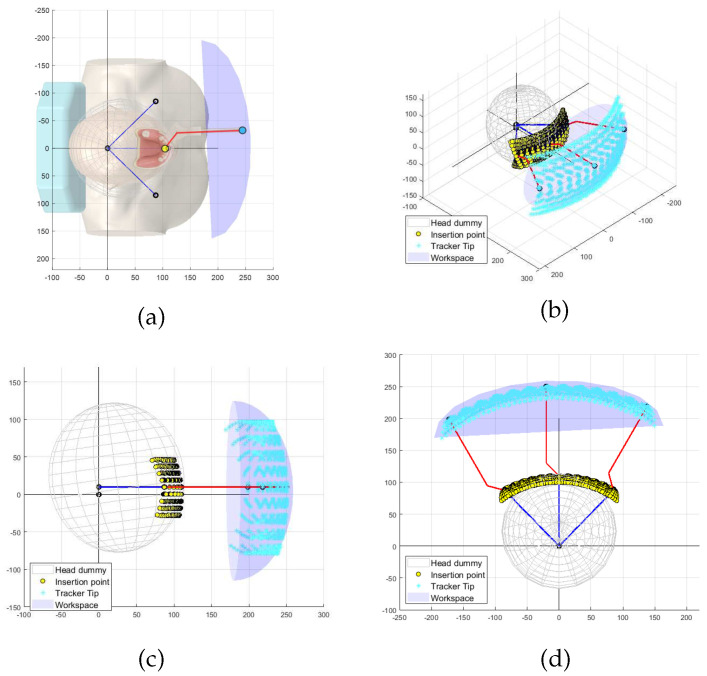
Simulated target workspace according to target head ROM: (**a**) schematic diagram of end-effector; (**b**) perspective view; (**c**) side view; (**d**) top view.

**Figure 4 sensors-23-02450-f004:**
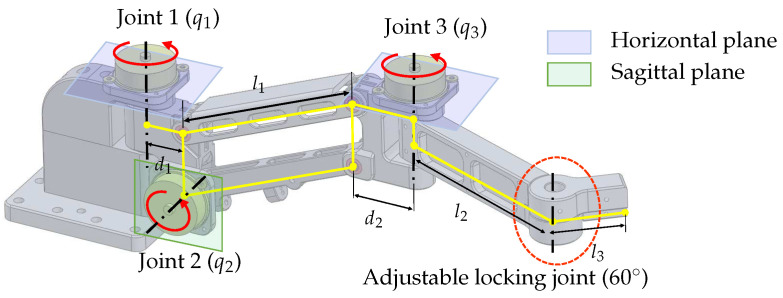
Configuration of the translational motion-tracking structure.

**Figure 5 sensors-23-02450-f005:**
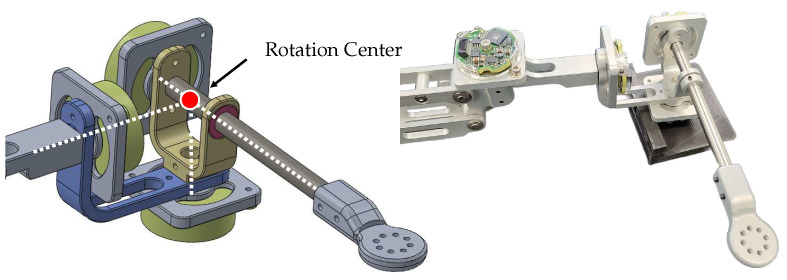
The rotational angle-tracking structure based on serial link mechanism.

**Figure 6 sensors-23-02450-f006:**
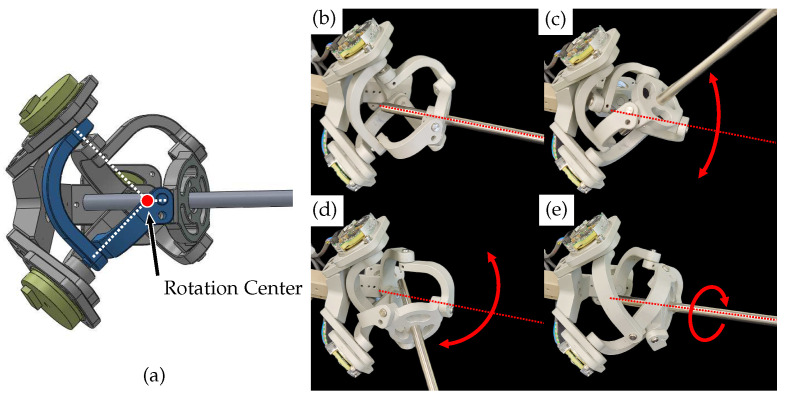
The rotational angle-tracking structure based on the Agile Eye mechanism: (**a**) linkage configuration of the one leg of the Agile Eye mechanism; (**b**) neutral posture; (**c**) pitch motion of the Agile Eye; (**d**) yaw motion of the Agile Eye; (**e**) roll motion of the Agile Eye.

**Figure 7 sensors-23-02450-f007:**
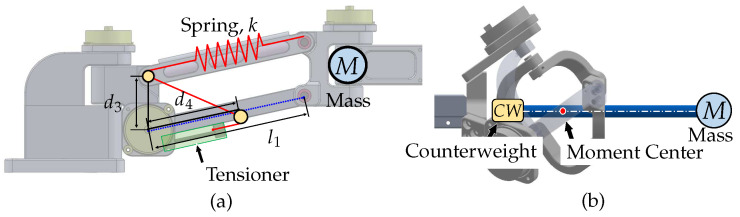
Gravity compensation mechanism for the motion tracker: (**a**) gravity compensation using spring–balancer mechanism; (**b**) gravity compensation using counterweight mechanism.

**Figure 8 sensors-23-02450-f008:**
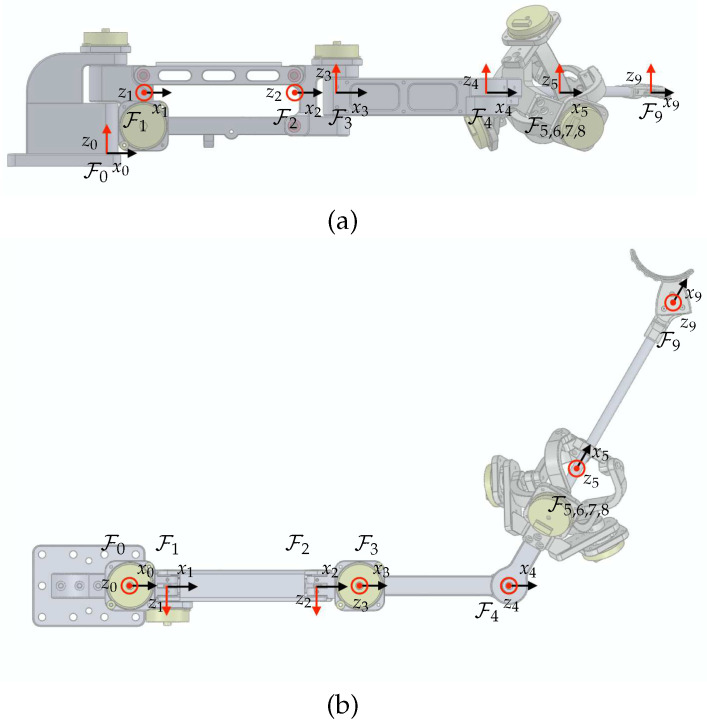
The frame assignments for the proposed link structure: (**a**) front view; (**b**) top view.

**Figure 9 sensors-23-02450-f009:**
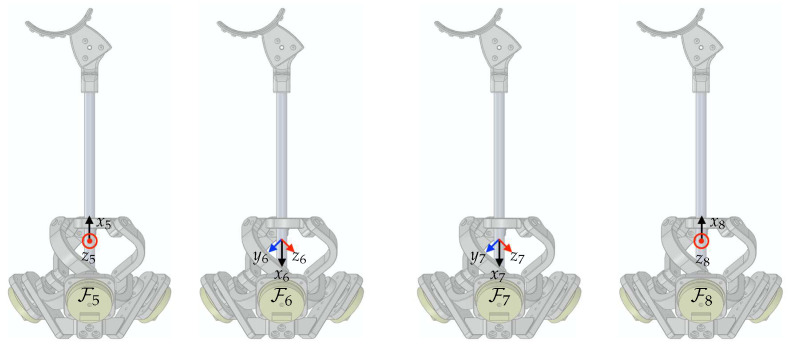
The frame assignments for the Agile Eye.

**Figure 10 sensors-23-02450-f010:**
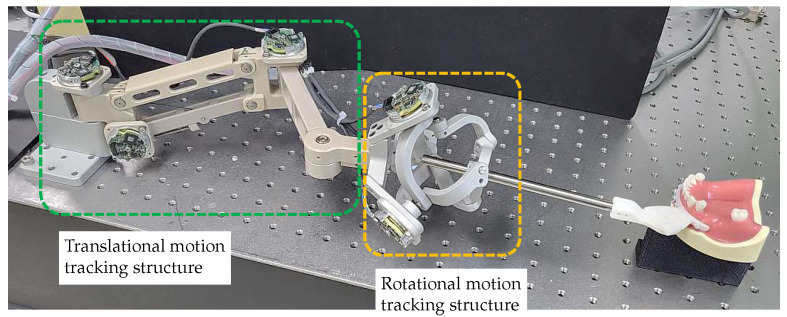
The lightweight motion-tracking arm for dental surgery.

**Figure 11 sensors-23-02450-f011:**
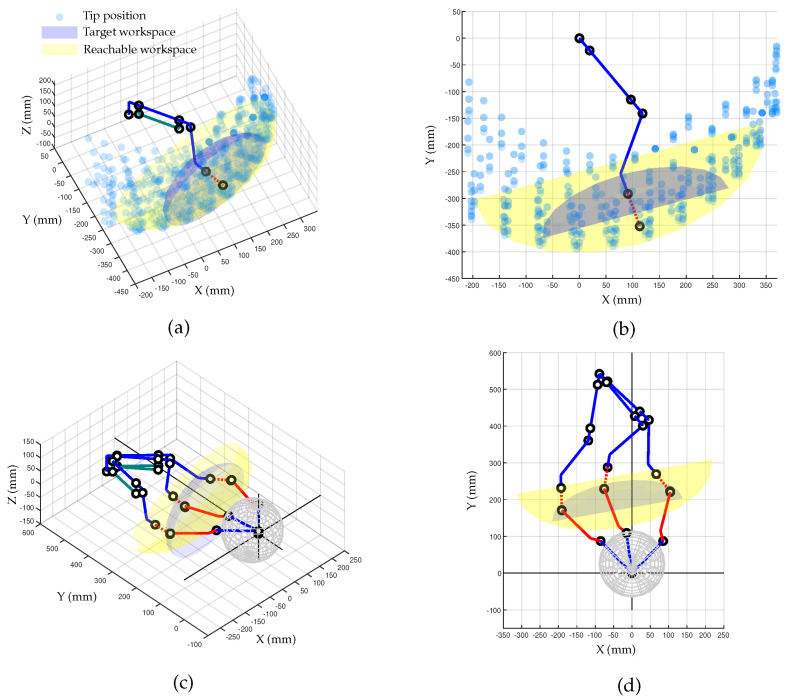
Reachable workspace computed via numerical simulation: (**a**) perpendicular view of the reachable workspace by the translational motion structure; (**b**) top view of the reachable workspace by the translational motion structure; (**c**) perpendicular view of the reachable workspace by the entire motion-tracking system; (**d**) top view of the reachable workspace by the entire motion-tracking system.

**Figure 12 sensors-23-02450-f012:**
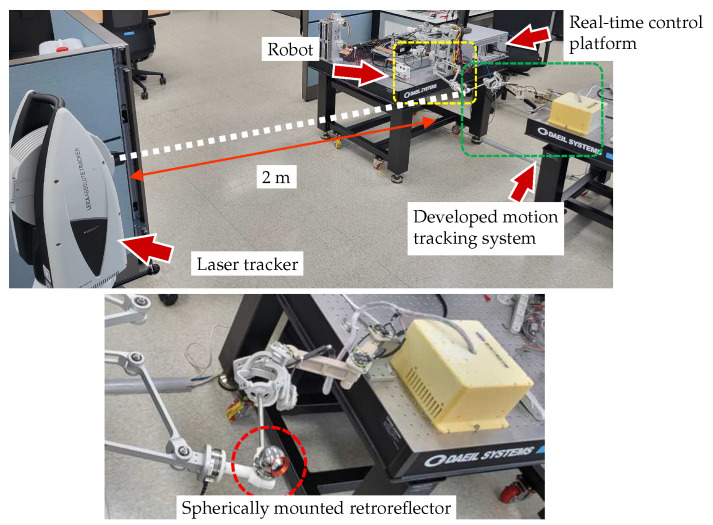
Experimental setup to evaluate the positional accuracy. A robot and real-time control platform were used to generate repetitive movement for the developed motion tracking system, and a laser tracker and a Spherical Mounted Retroreflector were used to measure position reference data.

**Figure 13 sensors-23-02450-f013:**
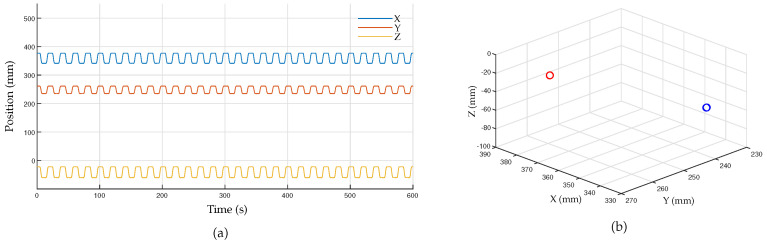
Experimental results on positional accuracy: (**a**) measured end-effector position from the developed motion-tracking system; (**b**) sampled points at the origin (red) and the target (blue).

**Figure 14 sensors-23-02450-f014:**
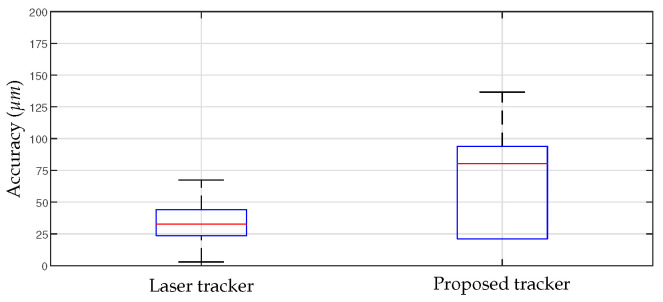
Experimental results on positional accuracy.

**Figure 15 sensors-23-02450-f015:**
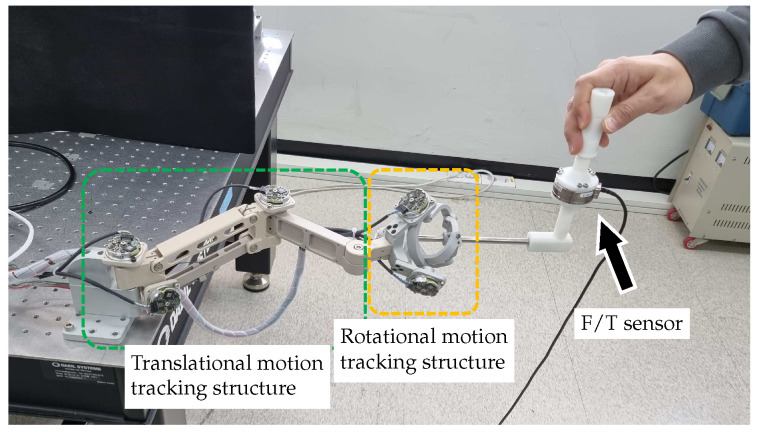
Experimental setup to evaluate back-drivability.

**Figure 16 sensors-23-02450-f016:**
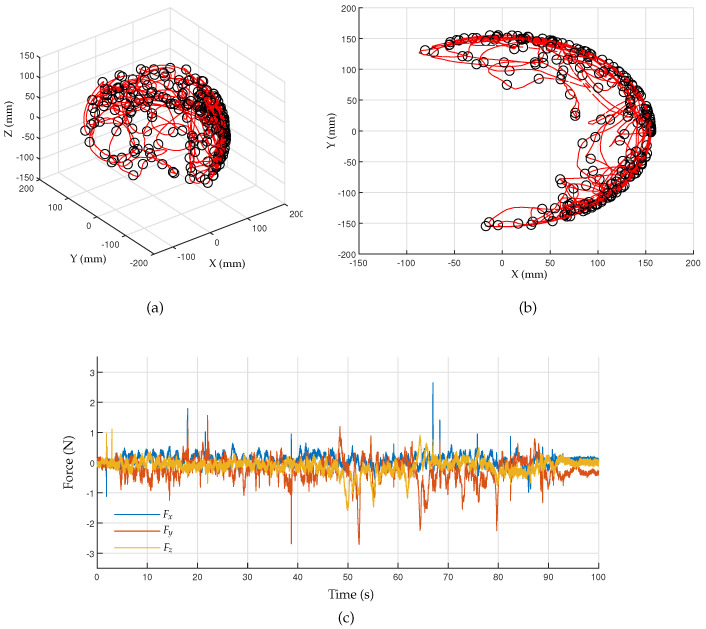
Experimental results for back-drivability: (**a**) trajectories of arbitrary motions generated by the operator; (**b**) top view of the trajectories; (**c**) force measured according to arbitrary human motions.

**Table 1 sensors-23-02450-t001:** Coronal and angular deviation of dental implant placement reported in the literature.

Method	Coronal Deviation [mm]	Angular Deviation [${}^{\circ }$]	Reference
Free-hand	1.62–1.93	5.85–8.7	[6,10,18]
Computer-guided implant surgical template	0.5–1.49	2.0–5.63	[4,6,18,19]
Optical tracker	0.55–1.25	0.89–3.24	[6,8,9,20]
YOMI	1.04	2.56	[21]

**Table 2 sensors-23-02450-t002:** Denavit–Hartenberg parameters for $F_{1}$ to $F_{5}$.

*i*	$q_{i}$	$\alpha_{i}$	$a_{i}$	$d_{i}$
0	0	0	0	0
1	$q_{1}$	$90^{\circ }$	$d_{1}$	$h_{1}$
2	$q_{2}$	0	$l_{1}$	0
3	–$q_{2}$	–$90^{\circ }$	$d_{2}$	0
4	$q_{3}$	0	$l_{2}$	0
5	$60^{\circ }$	0	$l_{3}$	0

## Data Availability

Not applicable.

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
