# Peer review of "Development of a Real-Time 6-DOF Motion-Tracking System for Robotic Computer-Assisted Implant Surgery"

_sensors, 2023, doi:10.3390/s23052450_

Round 1
Reviewer 1 Report
Section 2.1 "State of the Art" makes a description of the main components of a "motion tracking system" for implant surgery. They should review the state of the art, to include other similar "Robotic Computer-Assisted Implant Surgery" devices that have been developed by other researchers.
Section 5 "Results", must be completed with a comparative study of the results (accuracy, back-drivability), provided by the device developed in this paper, with the results provided by other similar "Robotic Computer-Assisted Implant Surgery" devices that have been developed by other researchers.
The English of the paper should be significantly improved.
Author Response
Thank you for taking the time to review our paper. We appreciate your constructive comments and suggestions. Please find the attached document for the response.

Reviewer 2 Report
1.The Manuscript writing style is more apt for a text book rather than a journal.
2.There are quite a few grammatical errors present throughout the manuscript which have been highlighed in the attached manuscript.
3.The application of their device would have been more credible if it was done realtime during implant surgery on a patient.
4.The limitations and scope of the study have not been discussed in the manuscript.
5.The manuscript can be trimmed quite a lot to hold the readers attention.

Author Response

(The authors gave the same response as above.)

Round 2
Reviewer 1 Report
With the response given by the authors, the paper has been accepted for publication.
Reviewer 2 Report
The corrections have been made according to the observations of the reviewers